# A Review of the Mechanisms and Risks of *Panax ginseng* in the Treatment of Alcohol Use Disorder

**DOI:** 10.3390/diseases13090285

**Published:** 2025-09-01

**Authors:** Eli Frazer, Candi Zhao, Jacky Lee, Jonathan Shaw, Charles Lai, Peter Bota, Tina Allee

**Affiliations:** 1School of Medicine, California University of Science and Medicine, Colton, CA 92324, USA; 2Psychiatry, College Medical Center, Long Beach, CA 90806, USA

**Keywords:** *Panax ginseng*, alcohol use disorder, traditional Chinese medicine, psychiatry, addiction

## Abstract

Alcohol use disorder (AUD) is a widespread, multifaceted disorder involving overproduction of pro-inflammatory cytokines, oxidative liver injury, and dysfunction of the brain’s dopaminergic reward circuits. Korean red ginseng (KRG), an herbal supplement derived from *Panax ginseng*, has demonstrated qualities potentially useful to the treatment of AUD, including antioxidative, anti-inflammatory, neuroprotective, and anxiolytic effects. This review examines active constituents of KRG, their pharmacological actions, and evidence supporting KRG’s therapeutic potential in the context of AUD, while also assessing its safety profile, adverse effects, and potential drug interactions. KRG’s main bioactive constituents, ginsenosides, appear to have roles in modulating alcohol-metabolizing enzymes, ethanol-activated inflammatory cytokine cascades, and neurological systems disrupted by AUD, including GABAergic and dopaminergic pathways. Evidence from animal models and limited small-scale human trials suggests KRG may alleviate symptoms of alcohol withdrawal, enhance cognitive performance, and attenuate anxiety through these pathways. While generally safe for consumption, several case reports and animal studies have indicated KRG’s potential to pose a variety of risks in vulnerable populations at high, prolonged doses, including hepatotoxicity, cardiovascular changes, mood disturbances, and hormonal effects. Furthermore, KRG’s neuromodulating role and influence on cytochrome P450 enzymes make it liable to interact with several medications, including warfarin, midazolam, selegiline, and serotonergic agents. Overall, KRG shows promise as a complementary supplement in managing aspects of AUD, though current evidence is limited by low sample sizes, inconsistent reports regarding nuances of ginsenosides’ mechanisms, and a low number of human trials. Further human-focused research is needed to elucidate its safety, efficacy, and mechanism.

## 1. Red Ginseng—Introduction

*Panax ginseng*, also known as Korean red ginseng (KRG) in its steamed and dried form, is one of over 14 species in the *Panax* genus [1]. Records of *Panax* species’ medicinal use date back over 5 millennia, with *P. ginseng* specifically being associated with the Manchurian Mountains in China [2]. The etymology of *Panax* lies in the Latin word “panacea,” which translates to “a remedy for everything,” reflecting its traditional usage across a broad range of ailments [2]. True to its name, KRG has continued to be cited in the modern era as being used in efforts to treat a wide variety of problems, including cardiovascular disease, central nervous system disorders, cancers, diabetes, inflammation, and addiction management [1,2,3,4,5,6,7]. This versatility can be reflected in its extensive global market, with a 2022 report estimating a worldwide production of 86,223 tons of fresh ginseng and an approximate global market value of $5.9 billion [3]. More broadly, herbal dietary supplements have become increasingly popular in the United States, with Americans spending approximately $9.6 billion on herbal supplements in 2019 alone, an 8.6% increase from the year before [4]. KRG’s wide accessibility can be attributed in part to the Dietary Supplement Health and Education Act (DSHEA) of 1994, which classified herbal supplements as subcategories of food, making them subject to far less pre-market oversight than conventional drugs [4]. This introduces challenges in assessing its safety, which is a crucial thing to know for patients susceptible to liver injury or drug interactions.

One such population is those with alcohol use disorder (AUD). AUD is a widespread concern, estimated to affect 28.9 million people over the age of 12 in the U.S. based on the 2023 National Survey on Drug Use and Health (NSDUH), and is estimated to be implicated in 3.8% of all global deaths [5,6]. Chronic alcohol use disrupts inhibitory and excitatory neurotransmitter signaling, most notably by downregulating gamma-aminobutyric acid (GABA) receptors and upregulating glutamate receptors, which contribute to withdrawal symptoms like anxiety, irritability, and seizures [7,8]. It also impairs the brain’s dopaminergic reward circuits, increasing cravings and relapse risk [8,9,10]. Beyond neuroadaptations, alcohol places enormous metabolic stress on the liver, especially through the upregulation of cytochrome P450 2E1 (CYP2E1) enzymes in the microsomal ethanol-oxidizing system (MEOS), which generate reactive oxygen species and accelerate hepatic injury [7,10,11]. Furthermore, alcohol consumption activates pro-inflammatory cytokines and signaling cascades—such as nuclear factor κ-light-chain-enhancer of activated B cells (NF-κB), cyclooxygenase 2 (COX-2), and inducible nitric oxide synthase (iNOS)—that contribute to both neurocognitive symptoms and alcoholic fatty liver disease (AFLD) [1,8,11]. For these reasons, anti-inflammatory drugs and supplements may be useful components in a regimen for treating alcohol use disorder.

Recently, interest has been growing in KRG’s potential to mitigate several of AUD’s pathophysiological processes. Ginsenosides, the primary bioactive components of KRG, have demonstrated anti-inflammatory, antioxidative, anxiolytic, and neuroprotective effects in many preclinical studies and some human trials [7,8,10]. While little research has been performed specifically on KRG’s capacity to impact alcohol cravings in AUD patients, there are a multitude of studies showing its impact on the symptoms of alcohol withdrawal and common comorbidities of AUD, like AFLD [2,3,4,6,7,12,13,14,15,16]. Similarly, several animal and human subject studies have cited KRG to improve the subjects’ overall well-being amidst a state of alcohol withdrawal [2,3,4,12]. Given this body of evidence and lack of centralized guidance on KRG’s side effects and interactions, a comprehensive review is warranted to assess the extent to which KRG’s pharmacological properties align with the therapeutic needs of individuals with AUD. This review seeks to examine the active constituents of KRG, their pharmacological actions, and the evidence supporting KRG’s therapeutic potential in the context of AUD. It will also assess the safety profile, adverse effects, and known drug interactions of KRG, highlighting both the promise and the limitations of this widely used but inconsistently regulated supplement.

To conduct this narrative review, the authors used various databases including PubMed, Scopus, Google Scholar, and Web of Sciences to find relevant literature. Some examples of search terms include the following: *Panax ginseng*, Korean red ginseng, ginsenosides, pharmacokinetics, alcohol use disorder, alcoholic fatty liver disease, oxidative stress, inflammation, traditional Chinese medicine, withdrawal, psychiatry, addiction, neuroprotection, side effects, and herbal supplements. Results of these search terms were individually checked for relevance and the titles within their reference lists were also examined for applicable references. The authors generally searched for peer-reviewed sources that addressed the proposed mechanisms of action of *Panax ginseng*, its side effect profiles, and any relevant clinical trials associated with this supplement. As this is a narrative review, no research protocol was registered.

## 2. Red Ginseng—Active Constituents and Mechanism of Action

Ginsenosides, ginseng’s primary bioactive constituents, are built on a steroid-like skeleton with a wide variety of side chains, differentiating them into over 180 documented types [1,7]. Ginsenosides are often classified as protopanaxadiols (PPDs), protopanaxatriols (PPTs), and oleanolic acid derivatives. PPDs Rb1, Rb2, Rc, and Rd, along with PPTs Re and Rg1, make up over 90% of the ginsenoside content in KRG roots [1,17]. When taken orally, the more hydrophilic ginsenosides of KRG (like Rb1, Rb2, and Rc) are converted to more hydrophobic variants (like compound K, Rg3, and F2) through a series of dehydration and glycosyl-cleaving reactions that are mediated by bacteria of the stomach and intestine [1,10,18,19]. These hydrophobic, more bioavailable PPDs are thought to be responsible for a wide variety of KRG’s pharmacological effects, which are discussed further in the therapeutics section below [7,10]. It is worth noting that the efficiency of this conversion varies significantly between individuals due to differences in gut microbiota composition, which can be shaped by diet, antibiotic use, and other host factors [10]. This variation can impact the ginsenosides’ pharmacokinetic parameters like absorption time and peak concentration, and in turn alter the clinical efficacy of KRG [10]. For a detailed depiction of ginsenoside bioconversion pathways, see the Introduction and Figure 1 of Kim et al.’s 2022 study on ginsenoside pharmacokinetics in human volunteers [10].

The versatility of ginsenosides’ therapeutic applications can be attributed to the compounds’ capacity to impact a plethora of cellular and biochemical mechanisms [1,8,12,13]. In the context of central nervous system (CNS) disorders, the literature provides evidence for ginsenosides’ roles in acetylcholine transmission (Rb1), inhibiting Tau phosphorylation (Rb1, Rg1, Rd), curbing glutamate toxicity (Rg2, Rb1), promoting neurite outgrowth and regeneration to improve neuroplasticity (Rb1 and Rg1), and decreasing amyloid-β induced inflammation (Re, Rg3, Rh2, Rb1) [8]. It is worth noting that there is overlap between ginsenosides like Rb1 and Rg3 in their anti-inflammatory mechanisms. Rb1 is documented more thoroughly in the literature for its neuroprotective properties, while Rg3 is more notable for its antitumor properties [8,14]. Regardless, because of ginseng’s many potential mechanisms and this review’s focus on its function in alcohol use disorder, this section will walk through the ways ginsenosides interact with the various pathways involved in alcohol consumption, from absorption in the GI tract to its long-term neurological effects.

### 2.1. Interference in Alcohol Absorption and Metabolism

When consumed orally, ethanol is absorbed through epithelial cells in the GI tract, then enters the bloodstream, and eventually gets transported to the liver. Here, ethanol is converted to acetaldehyde by alcohol dehydrogenase (ADH), which is then converted to acetic acid by aldehyde dehydrogenase (ALDH) [2,15]. However, this primary pathway becomes congested with frequent alcohol use, requiring the microsomal alcohol oxidation system (MEOS) to employ CYP2E1 isoenzymes to oxidize excess ethanol to acetaldehyde. This generates free radicals that damage hepatocytes, induce hangover symptoms, and cause the depletion of glutathione, which is needed to eliminate other pharmaceuticals [2,12].

Ginsenosides play a multifactorial role in combating these physiological changes induced by alcohol use, with a plethora of sources citing KRG extract’s ability to decrease ethanol absorption in the GI tract [7,16,20]. These all base their citation on a 1993 study that showed rats given ethanol orally had reduced plasma ethanol levels when co-administered red ginseng (21% lower of the area under the curve on a 0–5 h concentration vs. time plot) [21]. The study also showed ginseng had little impact on the blood ethanol levels when the alcohol was administered intraperitoneally, indicating it plays an important role in limiting absorption of ethanol, specifically from the GI tract [21].

More recent work suggests that ginseng may additionally speed up the metabolism of alcohol in the blood by promoting the expression of ADH and ALDH [2,22,23,24,25]. However, these effects might not be entirely due to the ginsenosides of ginseng. Several studies have shown that administering KRG extract with ethanol in rodents will decrease blood ethanol levels in comparison to the control groups for the hours following consumption, while also having relatively elevated blood acetaldehyde levels [22,23]. This would be consistent with ginseng upregulating the expression of ADH, but not ALDH. Interestingly, a 2014 study determined that ginsenoside-free molecules (GFMs), non-ginsenoside compounds obtained by steam-drying ginseng berries, were effective in reducing both ethanol and aldehyde levels in mice [25]. This was elaborated on in the same study with a set of in vitro experiments that determined the GFM linoleic acid was most effective in stimulating expression of aldehyde dehydrogenase, alcohol dehydrogenase, catalase, and CYP2E1, all essential enzymes in the metabolism of alcohol [25]. Furthermore, a 2021 study with mice found that administration of ethanol with injections from ginseng sprouts was able to reduce both ADH and ALDH gene expression levels, with coinciding decreased alcohol and acetaldehyde serum levels [26]. It is worth noting that ginseng sprouts contain a higher amount of GFMs compared to roots, which are much more concentrated with ginsenosides. Given that KRG extract is obtained from roots and consequently does not contain a high amount of these GFMs, it would make sense that aldehyde levels were not decreased in the studies where KRG extract was given if GFMs were responsible for the increase in the expression of ALDH [19]. However, there are studies that speculate on the potential of ginsenosides to increase ALDH gene expression [24]; it would likely be beneficial to carry out additional studies to clarify ginsenosides’ impact on this gene’s expression.

### 2.2. Anti-Inflammatory Effects

Outside of potentially modulating GI ethanol absorption and metabolic enzyme levels, ginseng is well-known for its robust anti-inflammatory effects [1,2,3,4,5,18,19]. After consuming alcohol, levels of several pro-inflammatory cytokines rise, including interleukin-6 (IL-6), IL-1β, and tumor necrosis factor α (TNF-α), which can activate the NF-κβ pathway, increasing nitric oxide and prostaglandin production via iNOS and COX-2 [2]. Chronic activation of these pathways can contribute to a plethora of pathologies, including various cancers, CNS symptoms associated with hangovers (e.g., headaches, drowsiness, memory impairment), and alcoholic fatty liver disease (AFLD) [1,2,12].

Luckily, ginsenosides have been found to inhibit several components of this inflammatory cascade. Rh1, Rh2, and compound K are known to suppress production of NO and PGE_2_ while inhibiting activation of NF-κβ [1]. Rb2, Rd, Rg1, and Re can inhibit lipopolysaccharide (LPS) induced nitric oxide (NO) formation and TNF-α production in microglial cells [1]. Rh2 (and Rg3 to a lesser extent) was shown to inhibit the production of NO and expression of COX-2, TNF-α, and IL-1β in microglial cells, while simultaneously increasing the expression of anti-inflammatory cytokine IL-10 [27].

The anti-inflammatory effects of these individual ginsenosides mirror KRG’s effects when given as an extract. A study with alcohol-treated mice found that co-administering KRG lowered levels of TNF-α and NF-κβ, while expressing higher levels of brain-derived neurotrophic factor (BDNF), a protein usually inhibited by TNF-α and NF-κβ that fosters neuronal growth and maturation [22]. These findings, in conjunction with the mice’s behavioral differences, align with past studies cited in this article that suggest neuroinflammation has a role in drug dependence, hinting that KRG may have a role in limiting these inflammatory processes [22]. KRG’s capacity to reduce inflammation through this pathway extends to its utility in combating features of AFLD, a common comorbidity of AUD. Ginseng was found across a review of several studies to suppress inflammatory cytokines TNF-α and IL-1β in both acute and chronic alcohol-induced liver injury models, protecting the liver from damage these cytokines can inflict through the pathways described above [15]. KRG was also found to enhance the expression of various antioxidant enzymes, including catalase, superoxide dismutase (SOD), glutathione peroxidase (GPx), and glutathione reductase in the liver, reducing the oxidative damage that contributes to alcoholic liver disease [15]. Collectively, these studies provide strong evidence for KRG’s anti-inflammatory properties, which extend its potential to treat both the acute and chronic consequences of heavy alcohol use.

### 2.3. Modulation of Neurotransmitter Systems

Following chronic alcohol exposure, the body downregulates the number or sensitivity of GABA receptors and increases the number or sensitivity of glutamate receptors in an effort to counter alcohol’s sedative effects [8]. When alcohol is removed from the body, the CNS and sympathetic nervous system (SNS) remain unbalanced in an overdrive state, resulting in the tremors, sweating, and tachycardia people experience in withdrawal or hangovers [12]. A 2005 study with mice found that ginsenosides Rb1, Rg1, and Rg5/Rk mixture had anxiolytic effects, and attributed this to the interaction observed between these ginsenosides and GABA receptors in other reports; however, the full anxiety-reducing mechanism behind this interaction remained unclear and was not thoroughly explored within this study [28]. Other rat-based studies cite a more complex role in anxiety and depression through suppression of corticotropin-releasing factor (CRF) expression and stimulation of neuropeptide Y (NPY) expression in the hypothalamus [13]. This incomplete framework, in which the remedial effect is demonstrated with only a partial understanding of how it is mediated, is consistent with current understanding of many of the ginsenosides’ therapeutic roles, including alcohol disorder. Furthermore, in the context of anxiety, depression, and substance use disorders, co-morbidities often associated with AUD, a plethora of sources cite ginsenosides’ interference with monoamine oxidase (MAO) enzymes and interactions with dopamine receptors to play an important role in helping treat these disorders [7,16,29,30]. Substance addictions are speculated to develop from alterations to neurotransmission and synaptic plasticity in structures of the mesolimbic pathway of the brain [3,4,5,13]. When an addictive substance like alcohol is used chronically, more and more dopaminergic neurons are stimulated to the point where dopamine supply in the mesolimbic reward circuits can become depleted; this depletion causes both increased cravings for the substance and more severe withdrawal symptoms [7]. Ginsengs have been cited to have use in treating a plethora of addictions (including cocaine, opiates, psychostimulants, and nicotine) through its modulation of dopaminergic pathways [3,4,5,6]. Considering this property, along with KRG showing the capacity to restore functionality to the dopaminergic mesolimbic pathway in this study on anxiety in rats with ethanol withdrawal (EW), there is evidence that KRG has the potential to combat the addictive component of alcohol use disorder, something that is elaborated on further in the therapeutics section.

### 2.4. Additional Potential Mechanisms in Long-Term Alcohol Use

Another study carried out with rats induced with AFLD demonstrated that coadministration of KRG resulted in partial reversal of the alcohol-induced suppression of AMP-activated protein kinase (AMPK) phosphorylation and consequent upregulation of acetyl-CoA carboxylase activity in the liver [31]. Chronic alcohol use impairs AMPK activation and consequently disrupts the subsequent phosphorylation of acetyl-CoA carboxylase (ACC) by AMPK. ACC carboxylates acetyl-CoA (acetyl-coenzyme A) into malonyl-CoA, with high malonyl-CoA levels suppressing fatty acid oxidation while promoting lipid synthesis. Thus, when AMPK is inhibited, more ACC is able to remain active and produce malonyl-CoA, leading to overall increased fatty acid build-up while reducing its breakdown [31]. By reversing this alcohol-induced inhibition and consequently combating hepatic fat accumulation, KRG shows the capacity to counteract one of the most pathogenic factors of AFLD. Furthermore, KRG was found in a 2023 meta-analysis to reduce serum triglycerides, low-density lipoprotein (LDL), and hepatic cholesterol levels across the seven relevant animal studies examined, providing more evidence for its potential utility in managing AFLD [15].

Additionally, some of the anti-inflammatory properties of ginsenosides are linked to their immunomodulatory effects [1]. Immune responses are controlled in part by T helper (Th) cells, with Th1 cells and macrophages mediating the cell-mediated part and Th2 cells carrying out the humoral (antibody-mediated) part [1]. Th cells activate and direct other immune cells like cytotoxic T cells and NK cells, making them central to the immune process. Th1 cells are activated by dendritic cells (DCs) that present antigens and secrete type 1 cytokines (IL-2, interferon-γ [IFN-γ], and IL-12), while Th2 cells are activated by DCs that present antigens and generate type 2 cytokines (IL-4, IL-10) to help B cells secrete protective antibodies. Therefore, compounds that can regulate these mechanisms have the potential to significantly impact chronic inflammation. PPT-type ginsenosides F1 and Rg1 have been found to influence type 2 cytokine production by regulating expression of IL-4, while Rh1 was shown to influence type 1 cytokine production through regulation of IL-12 and IFN-γ expression [1]. Another study found Rb1, Rb2, Re, and Rg1 influenced B and T lymphocyte proliferation induced by regulating expression of T-cell mitogens, B-cell mitogens, LPS, and IL-2 while simultaneously suppressing production of inflammatory cytokines IL-6 and IL-1β [1]. Furthermore, DC maturation has been shown to be promoted by ginsenosides like compound K, as one study showed enhanced differentiation of naive T cells towards Th1 type if the DC could secrete IL-12 [1]. A broad overview of the potential mechanisms of action for *Panax ginseng* in AUD can be seen in Figure 1.

## 3. Red Ginseng—Therapeutic Effects

KRG has been cited to have a role in treating a wide range of ailments, including cardiovascular disease, central nervous system disorders, cancers, diabetes, inflammation, and addiction management [1,2,3,4,5,6,7]. In the context of AUD, KRG’s therapeutic effects can be found in the way it treats the user’s craving for alcohol, withdrawal symptoms after alcohol use, and comorbidities associated with AUD. While some may argue that mitigating the symptoms of withdrawal can serve as a means to encourage alcoholics to consume the beverage more frequently, it is important to note that withdrawal symptoms are often a prominent factor in preventing patients from quitting alcohol in the first place. Though little research has been conducted specifically on KRG’s capacity to impact alcohol cravings in AUD patients, there are a multitude of studies showing its impact on the symptoms of alcohol withdrawal and common comorbidities of AUD like alcoholic fatty liver disease. From a general perspective, a plethora of sources have cited KRG’s use in the management of chronic alcoholism [2,3,4,6,7,12,13,14,15,16]. Similarly, several animal and human subject studies, as seen in Table 1, have cited KRG to improve the subjects’ overall well-being amidst a state of alcohol withdrawal [2,3,4,12].

### 3.1. Symptom Relief in Ethanol-Induced Hangovers—A Human Study

A 2014 study [23] evaluated KRG’s impact on the symptoms of ethanol-induced hangovers in 25 male humans over the course of two weeks. During the first visit, patients received 100 mL of whiskey followed by either 100 mL of a water placebo or 100 mL of a drink containing KRG extract. Patients had their blood drawn at intervals from 0 to 240 min after consuming these, and were asked to complete a survey regarding their symptoms once they returned home 24 h from the time of consumption. A week after the first visit, the whole process was repeated with the placebo and KRG groups switched. A week after the second visit, patients were given alcohol once more and asked to fill out the survey a third time as a positive control trial, with neither group receiving a placebo or KRG extract. Of the 15 symptoms inquired from the survey, patients reported improved experiences with memory impairment, concentration, feelings of thirst, fatigue, and stomachaches when receiving KRG extract with ethanol, compared to receiving ethanol with a placebo. In addition, participants reported overall improvement to the withdrawal symptoms after visits in which they received the KRG extract. Little difference between the KRG and placebo conditions was seen for the other withdrawal symptoms in the survey, which included tremors, headache, nausea, vomiting, diaphoresis, feelings of anxiety, feelings of depression, or dizziness. It is worth noting that on the questionnaire’s severity scale of 0–4, the average rating for all of these symptoms even in the positive control state did not exceed 1.00 (mild symptoms), with the exception of headache, dizziness, fatigue, concentration, feelings of thirst, and stomachaches (the latter four of which were improved significantly with KRG extract). In comparison to the groups receiving alcohol with a placebo, participants receiving KRG and alcohol had lower blood alcohol levels for time points between 30 and 60 min after consuming the drinks, and a significant increase in acetaldehyde levels at the time points between 30 and 240 min. This is interesting considering acetaldehyde is commonly reported to be correlated to hangover symptom severity; one study showed that mice with a knockout gene for aldehyde dehydrogenase showed more severe symptoms of hangover [2,3,4,12,26]. This study does point out that there are limitations for these results since each participant only completed one symptom survey for each visit. These surveys were taken 24 h after consuming alcohol and KRG, which was likely several hours after the severity of hangover symptoms had peaked. A potentially more accurate way to gauge the participants’ conditions would have been to give surveys prior to taking the drinks, 2 to 4 h after taking the drinks, and 24 h after taking the drinks [23].

### 3.2. Effects on Alcohol Withdrawal in Animal Models

Unfortunately, few subsequent projects on KRG’s impact on AUD have been carried out with human subjects. However, there have been a multitude of experiments with animal subjects that provide additional evidence for KRG’s utility in this area. A 2023 study compared the behaviors of ethanol-intoxicated mice with various levels of KRG extract pre-treatment [22]. After being administered alcohol and KRG, the mice were observed for subsequent behaviors reflecting withdrawal symptoms, including jumping, rearing, facial washing, shaking, forepaw tremors, and scratching. The data reflected significantly lower frequencies of the first four behaviors in the KRG groups compared to the ethanol-only control group, with no significant difference being identified in forepaw tremor or scratching frequency. Overall withdrawal scores were significantly lower for the KRG groups compared to the ethanol-only control group. Further tests with a Y-maze, Barnes Maze, and Novel Object Recognition Test showed mice given ethanol performed better if given KRG just prior, supporting its potential to improve spatial working memory when impaired by alcohol. The study examined the biochemical impact of KRG on alcohol metabolism as well, with KRG groups having lower blood alcohol levels and higher blood acetaldehyde levels than the ethanol control group; this aligns with the results of the previously discussed study. As mentioned in the bioactive compounds section above, additional experiments with the mice were carried out that showed KRG’s potential as an anti-inflammatory agent, with specific potential for neuroprotection based on its upregulation of BDNF [8]. When considering this with the mice’s improved symptoms, it is possible that the benefits of KRG’s anti-inflammatory propensity may outweigh the uncomfortable effects of the increased blood acetaldehyde levels it induces [22].

A 2014 study [29] explored the KRG extract’s effect on rats experiencing ethanol withdrawal (EW), focusing more on its anxiolytic effects mediated by the dopaminergic pathway in the central nucleus of the amygdala (CeA). Dopamine deficiency in the central nucleus of the amygdala has an established correlation with anxiety, suspected to be a result of reduced neuron firing in the ventral tegmental area (VTA). Given KRG’s suspected role in modulating addictions to other substances through the dopaminergic pathway, the researchers in the study sought to explore this hypothesis [4,5,6,13]. Rats were treated with ethanol over the course of 28 days and were stopped to induce a state of withdrawal for 3 days. Following this, the rats’ level of anxiety was assessed with an elevated plus maze, which compares the amount of time spent in closed and open rooms of the maze. Following this test, the rats’ blood was drawn for plasma corticosterone levels, since this is an established marker of anxiety in rats. Results indicated that the rats in a state of EW spent significantly less time in open areas of the maze compared to the negative control group, suggesting a higher level of anxiety induced by alcohol. The rats that received KRG extract spent more time in the open areas of the maze, suggesting it may have anxiolytic effects. This notion is further supported by the plasma corticosterone levels being lower in KRG-extract-treated rats. A follow-up experiment in which rats were administered a dopamine D1 receptor (D1R) antagonist, D2R antagonist, or lactate ringers as a control. The D2R antagonist group showed increased anxiety while the D1R antagonist group showed reduced anxiety, indicating that D2 receptors may play a role in mediating KRG’s anxiolytic effects in the context of EW [29].

These results are supported by a series of prior animal studies [29,30,31]. One showed red and white ginseng had a comparable effect to diazepam on mice’s anxious behaviors when given orally over the course of several days, with associated inhibition of MAO activity [30]. Another showed Rb1 diminished the effects of separation anxiety in sixty-four 5-day old chicks [32]. Finally, another study with mice in elevated plus mazes found Rb1, Rg1, and Rg5/Rk mixture showed anti-anxiety effects comparable to diazepam, but with diminished locomotor side effects than diazepam; the anxiolytic effects of these ginsenosides were suspected to be a result of their interactions with GABA receptors observed in the same experiment [28]. Interestingly, KRG’s apparent anxiolytic capacity in rats runs in contrast to the aforementioned 2014 human trial study that found KRG had no significant impact on EW-induced anxiety; however, the self-rated average levels of anxiety in the latter study never exceeded 0.16 on a scale of 0–4, suggesting KRG likely had little capacity to express its anxiolytic effects in that scenario [23,29]. Likewise, that same study showed a similar lack of noticeable change in feelings of depression, though a plethora of animal studies show ginsenosides have some therapeutic potential in combating such symptoms. One study was performed with an intestinal metabolite of KRG, 20 (S)-protopanaxadiol, which showed comparable antidepressant effects to fluoxetine by increasing monoamine neurotransmitters with an anti-reuptake effect throughout the brains of mice [33]. The abundance of animal studies related to KRG’s potential antidepressive and anxiolytic effects and relative scarcity of thorough human studies on the subject indicate the necessity of carrying out more of the latter to better evaluate KRG’s clinical potential in these areas.

### 3.3. Modulation of Addiction-Related Neurocircuitry

It is worth noting that substance addictions are speculated to develop from alterations to neurotransmission and synaptic plasticity in structures of the mesolimbic pathway of the brain, the same area of the brain affected in the previously mentioned 2014 study [3,4,5,13]. When an addictive substance like alcohol is used chronically, more and more dopaminergic neurons are stimulated to the point where dopamine supply in the mesolimbic reward circuits can become depleted; this depletion causes both increased cravings for the substance and more severe withdrawal symptoms. Thus, a common goal in therapy for substance abuse is restoring normal dopamine function in these circuits [7]. Ginsengs have been cited to have use in treating a plethora of addictions (including cocaine, opiates, psychostimulants, and nicotine) through its modulation of dopaminergic pathways [4,5,6]. Considering this property along with KRG showing the capacity to restore functionality to the dopaminergic mesolimbic pathway in this study on anxiety in rats with EW, there is evidence that KRG has the potential to combat the addictive component of alcohol use disorder. For an overview of how KRG combats the neural effects of chronic alcohol consumption, see Figure 2 below.

### 3.4. Protective Effects Against Alcoholic Fatty Liver Disease

A frequent comorbidity of AUD is alcoholic fatty liver disease (AFLD). KRG has been shown in many studies to have strong potential for use in counteracting some of the disorder’s most deleterious effects, including downregulation of inflammatory cascades, control of oxidative damage, and minimization of fat accumulation in the liver [15,31]. See the last two paragraphs of the active metabolites section for a more thorough explanation of the mechanisms behind these properties.

Taken together, these findings suggest that KRG may offer several therapeutic benefits in the context of AUD, particularly in mitigating withdrawal symptoms, supporting cognitive function, and attenuating common comorbidities such as anxiety, depression, and AFLD. However, while animal studies provide strong foundational evidence, the limited number of rigorous human trials underscores the need for further clinical validation before KRG can be recommended in AUD treatment protocols.

## 4. Red Ginseng—Side Effects

KRG’s safety profile remains an area of ongoing debate due to conflicting clinical data stemming from a low number of trials and different preparation techniques of extracts throughout studies. The side effects reported throughout the literature span hepatic, cardiovascular, psychiatric, endocrine, and gynecological systems, though many findings are either of minimal clinical significance or in conflict with the results of similar studies.

### 4.1. Hepatic Concerns

Herb-induced liver injury (HILI) is among the most significant risks associated with herbal and dietary supplements, particularly in individuals with pre-existing liver damage, including those with alcohol use disorder [4,55,56]. The type of damage usually inflicted by herbal supplements tends to be idiosyncratic rather than intrinsic, meaning the injuries occur infrequently and in a somewhat unpredictable, dose-independent fashion [57]. The sporadic nature of these injuries makes examining the hepatotoxicity of herbs like *Panax ginseng* difficult [57]. Most of the literature on HILI focuses on other herbs and ginseng species [4,55,56], though there is at least one case implicating *Panax ginseng* in hepatotoxicity via drug–herb interaction that is recounted in a review of ginseng misuse cases [34]. In a patient on long-term imatinib therapy, concurrent use of ginseng energy drinks was temporally associated with acute lobular hepatitis, possibly due to CYP3A4 inhibition by ginseng leading to elevated plasma levels of imatinib [34,35]. However, the review notes that several other sources indicate ginseng’s propensity to induce rather than inhibit CYP3A enzymes, leaving the hepatotoxic potential of red ginseng specifically somewhat inconclusive [34].

### 4.2. Cardiovascular Effects

Animal and clinical studies offer mixed insights into *Panax ginseng*’s cardiac effects. A 2023 study found that ginsenoside-rich red ginseng did not alter electrocardiogram (EKG) parameters or cardiac histology in mice, though it did produce elevated creatine kinase levels and impaired calcium handling, indicating potential risk of subclinical myocardial stress [36]. Other animal studies cited within this work suggest that high doses or prolonged use of red ginseng may contribute to diastolic dysfunction and could predispose a person to heart failure with preserved ejection fraction if taken at an unspecified high dosage for an extended period of time [37]. In contrast, another study cited in the article using guinea pig hearts highlighted some of ginseng’s cardioprotective effects, with researchers reporting significantly reduced cardiac biomarkers like creatine kinase MB (CK-MB) and troponin I, lower oxidative stress markers, and better modulated inflammatory cytokines [38].

In humans, a study involving 30 healthy adults showed a statistically significant, albeit clinically minor, QTc prolongation and a modest drop in diastolic blood pressure after ginseng ingestion [34]. More concerning was a case report of a 43-year-old woman who consumed up to 4 L of ginseng per day, and subsequently presented with syncope and extreme QT prolongation, although excessive caffeine intake may have confounded this presentation [39]. These findings suggest that while KRG may confer cardioprotective effects under controlled conditions, excessive or prolonged use may pose cardiac risks in susceptible individuals or when combined with other stimulants.

### 4.3. Neuropsychiatric Effects

Several case reports have linked ginseng use to acute manic episodes [34,40]. A duo of reviews detail 4 separate cases of individuals experiencing manic episodes after starting consumption of large amounts of ginseng for several weeks, with resolution of symptoms after cessation of ginseng [34,40]. Only one of the case’s manic episodes can be attributed to a herb–drug interaction, where a 56-year-old woman took haloperidol and clomipramine in tandem with the ginseng [34]. Though most users do not experience psychiatric effects, the findings of these reports do raise questions about KRG’s potential to trigger mood dysregulation on its own in rare cases, especially with the knowledge of ginsenosides’ capacity to interact with a variety of neurotransmitter pathways. 

### 4.4. Endocrine and Gynecologic Effects

The structural similarity between ginsenosides and steroid hormones like estradiol has raised some concerns about ginseng’s potential endocrine-disrupting effects. A pair of case reports describe gynecomastia in a man who had ingested ginseng for a long period of time, and a 12-year-old boy who experienced gynecomastia after consuming ginseng for a month, with the gynecomastia resolving after discontinuing the boy’s ginseng prescription [34]. Another case report discussed a 39-year-old female experiencing menometrorrhagia and tachycardia after starting to use topical ginseng for cosmetic reasons; symptoms ceased after discontinuation of ginseng [41]. To contrast, a small clinical trial involving young women found that KRG reduced urinary bisphenol A (BPA) levels, an endocrine disruptor, leading to improvement of symptoms like menstrual pain and irregularity [42]. The underlying mechanism behind all these cases remains inconclusive, with findings in the literature showing certain ginsenosides’ capacity to activate estrogen α and β receptors in vitro, but not in vivo [34]. Likewise, no estrogen-like activity was observed when ginseng extract was administered to Ishikawa cells, a human endometrial adenocarcinoma cell line with an estrogen-sensitive alkaline phosphatase enzyme [34]. These findings necessitate further research to elucidate the ginsenosides’ mechanisms in different contexts [34].

Taking all these potential effects into consideration, KRG generally seems to be regarded as safe when used appropriately, but several reports of hepatic, cardiac, psychiatric, and endocrine effects, especially in high doses or vulnerable populations, warrant caution. The conflicting data across studies also highlights the importance of standardizing ginseng preparations and conducting more large-scale, controlled trials to clarify its risk profile.

### 4.5. Collective Side Effect Profile

There appears to be a limited number of clinical trials designed specifically to evaluate the side effects of *Panax ginseng*, with most safety concerns instead reported in case studies as seen in Table 2 below. Across the past 46 years, we identified an explicit total of 170 patients across 9 case reports and 2 clinical trials with documented side effects (unrelated to drug interactions) potentially linked to *Panax ginseng*. Studies examining the hepatotoxic effects of ginseng tend to group ginseng with herbal supplements as a whole [4,56]. The existing data suggests that the overall prevalence of severe adverse reactions to ginseng is relatively low. However, this perceived prevalence may be artificially lower due to patients potentially not reporting their herbal medicine use as well as under recognition/diagnosis of the side effects [4]. This is further complicated due to differences in ginseng products, which vary in ginsenoside composition and dosage [56]. Additionally, many of these findings are correlative and often based on diagnoses of exclusion [4]. Numbers needed to harm or effect sizes were provided in the literature reviewed, so the clinical relevance of the side effects of *Panax ginseng* are difficult to conclude from the existing literature.

## 5. Red Ginseng—Drug Interactions

*Panax ginseng* and its ginsenosides have been reported to interact with a variety of commonly prescribed medications through different mechanisms. These interactions are mediated through a variety of pathways, including cytochrome P450 enzymes, P-glycoprotein (P-gp), organic anion-transporting polypeptides (OATPs), central nervous system targets such as MAO and monoamine transporters, and cardiovascular mechanisms that influence platelet aggregation and vascular tone [35,43,44,58,59]. The clinical significance of these interactions suggested by the literature varies considerably between sources, with many only occurring at high ginseng dosages and others occurring with inconsistent results across studies.

### 5.1. Serotonin Syndrome

Serotonin syndrome describes a severe drug reaction characterized by autonomic hyperactivity, neuromuscular abnormalities, and mental status changes as a result of a person taking multiple agents that raise serotonin levels [60]. The condition can manifest in many different ways, with most cases involving tachycardia, diarrhea, shivering, diaphoresis, mydriasis, tremor, twitching, and hyperreflexia, and more severe cases involving severe swings in blood pressure from hypertension to shock, muscle rigidity, hyperthermia, rhabdomyolysis, and metabolic acidosis [60]. Several types of serotonin receptors are thought to collectively contribute to serotonin syndrome’s manifestation when activated, including the 5-hydroxytryptamine 2A (5-HT_2A_) receptor that ginsenosides like Rb1 have been found to use to carry out their antidepressant-like effects [61]. A 2017 study with mice showed that ginsenoside Rb1 significantly increased the amount of head twitching, suspected to be a result of its serotonergic effects [45]. A 2019 study with rats showed white ginseng (*Panax ginseng Meyer*) could raise serotonin levels in the hippocampus [46]. Due to these serotonergic effects, *Panax ginseng* can contribute to the eponymous syndrome, making it liable to interact with a plethora of other serotonergic herbs (such as St. John’s wort [60]) and a wide variety of SSRIs, SNRIs, atypical antidepressants, analgesics, and antibiotics that raise serotonin levels [60]. On the other hand, some constituents of ginseng, like ginsenoside Re, have been found in mice to inhibit 5-HT_2A_ receptors, reducing some of the serotonergic effects of other drugs [47]. Thus, some ginsenosides of *Panax ginseng* seem to have opposing serotonergic effects, despite the herb overall appearing to contribute more to elevated serotonin levels.

### 5.2. Warfarin and Antiplatelets

Despite being widely researched, the nature of ginseng’s interaction with warfarin, a vitamin K antagonist used widely as an anticoagulant, remains inconclusive and controversial [43]. The interaction between ginseng and warfarin has produced conflicting evidence in both case reports and clinical studies. A frequently cited 1997 case report documented a 47-year-old patient on a stable warfarin regimen whose INR dropped from 3.0–4.0 to 1.5 within two weeks of starting *Panax ginseng* (referred to as oriental ginseng in the report), without any other changes to his medications, lifestyle, or diet [48]. After discontinuing ginseng, the patient’s INR returned to baseline, suggesting a reduction in warfarin efficacy potentially linked to ginseng intake, although the relevant mechanism was stated to be unclear at the time [48]. This antagonistic effect was later supported by a 2004 randomized controlled trial in which 20 healthy volunteers were given either American ginseng or a placebo during weeks 2 to 3 of a staggered 4-week warfarin regimen [43]. Compared to the group receiving the placebo, those receiving ginseng exhibited significantly lower peak international normalized ratio (INR) and plasma warfarin concentrations, indicating that American ginseng reduced warfarin’s anticoagulant effect. While the authors suggested that the ginsenosides may have increased warfarin’s metabolism or clearance through interactions with hepatic enzymes, no specific mechanism was explored or proposed [43].

However, not all studies have supported this interaction. A 2008 two-week randomized controlled clinical trial of 25 ischemic stroke patients receiving warfarin either with or without *Panax ginseng* showed no statistically significant differences in INR, peak prothrombin time (PT), or area under the curve (AUC) between the two groups, even though all patients showed increased anticoagulation relative to baseline [44]. This suggests that *Panax ginseng* may not consistently antagonize warfarin, or that its impact could depend on unclear patient-specific factors, ginseng species, preparation method, or dosing regimens. Despite overall ginseng extract dosing being similar between the 2004 and 2008 studies, the former employed American ginseng while the latter used *Panax ginseng*, with the former having higher constituent proportions of Rb1 (1.93% vs. 0.20%) and Rg1 (0.35% vs. 0.10%) than the latter based on preparation details from the two studies [43,44]. Furthermore, a 2019 review supports the ambiguity in the interaction [3]. It was noted from the four ginseng-related studies it reviewed that, while American ginseng demonstrated antagonism in the aforementioned 2004 study, the other trials involving *Panax ginseng* or *Panax notoginseng* showed little to no effect on warfarin anticoagulation [59].

Beyond its potential warfarin interactions, the same 2019 review suggested that ginseng exhibits potential interactions with aspirin, a widely used antiplatelet drug [59]. A duo of preclinical pharmacokinetic studies [49,50] with rats was cited. The first revealed that co-administration of *Panax notoginseng* and aspirin significantly increases plasma salicylate concentrations, roughly doubling them compared to aspirin alone [49]. The second study showed increased intestinal permeability of both aspirin and salicylate with co-administration of ginseng, suspected to be mediated by ginsenoside-induced changes in membrane fluidity and tight junction integrity [50]. Interestingly, aspirin itself also enhanced the absorption of ginsenosides such as R1, Rg1, Rb1, Re, and Rd, suggesting a mutual enhancement of absorption when the two agents are used together [50]. While these findings suggest a plausible pharmacokinetic interaction, clinical evidence remains limited, and further research is needed to assess the significance of these interactions in patient populations at risk for bleeding.

### 5.3. Neuropsychiatric Agents: Selegiline, Phenelzine, and Midazolam

One 2020 rat-based study found *Panax ginseng* demonstrated a dose-dependent, biphasic interaction with selegiline, a MAO-B inhibitor often prescribed for Parkinson’s disease, with the oral bioavailability of selegiline being significantly altered by ginseng extract administration [51]. At baseline, selegiline’s bioavailability was approximately 18%. This dropped to 7.2% with low-dose ginseng but increased markedly to 29% with high-dose ginseng administration. These opposing effects appear to be mediated by cytochrome P450 enzymes, particularly CYP3A4, CYP2B6, and CYP1A2, which metabolize selegiline into desmethyl-selegiline and L-methamphetamine [51]. While ginseng is known to induce CYP1A2, high-dose ginseng was also found to inhibit CYP3A4, an enzyme crucial to both oxidative pathways of selegiline metabolism. Since CYP3A4 participates in a broader range of metabolic reactions than CYP1A2, its inhibition likely overrides the inducing effect on CYP1A2, resulting in slower metabolism and higher systemic levels of selegiline [51]. This interaction would have interesting clinical implications if verified with further studies. At lower ginseng doses, patients may experience reduced selegiline exposure, potentially diminishing therapeutic efficacy. At higher doses, increased selegiline bioavailability may raise the risk of adverse effects in the form of serotonergic symptoms or sympathomimetic toxicity [51]. If extrapolated to humans, these findings underscore the need for careful monitoring and dose adjustment when ginseng is co-administered with selegiline or other MAO inhibitors.

*Panax ginseng* may also influence the metabolism of midazolam through modulation of cytochrome P450 CYP3A enzymes. A 2012 controlled pharmacokinetic study investigated this by administering the CYP3A substrate midazolam and the P-glycoprotein substrate fexofenadine to 12 healthy participants before and after 28 days of *Panax ginseng* supplementation (500 mg twice daily) [52]. The study found that *Panax ginseng* significantly reduced midazolam levels systemically, with the medication’s AUC, half-life, and peak plasma concentration all decreasing with ginseng exposure. The authors suggest from these findings that *Panax ginseng* likely induces CYP3A activity in both the liver and gastrointestinal tract, thereby increasing the metabolism of CYP3A substrates such as midazolam [52]. Interestingly, this contrasts with other evidence from the literature that has found ginseng to inhibit CYP3A enzyme activity, as discussed in the side effects section [34,51]. Furthermore, another study with 14 healthy male participants sought to clarify *Panax ginseng*’s impact on cytochrome P450 enzymes’ activity on a variety of drugs and found slight inhibition of CYP3A4 when patients were given midazolam with ginseng [53]. Thus, it seems more research is required to better determine ginseng and its constituents’ effects on midazolam metabolism and CYP3A4 enzyme activity as a whole.

An interaction that has been less extensively studied is that between *Panax ginseng* and phenelzine, an MAO inhibitor used as an antidepressant [54]. A single, frequently cited case report from 1987 provided mild evidence that the herb’s extracts could enhance the antidepressant’s efficacy [54]. The case report in question followed a 42-year-old woman with a history of depression who was started on phenelzine at a gradually increasing dose up to 45 mg daily. She was taking no other medications at the time with the exception of “ginseng and bee pollen” [54]. Soon after starting the medication, she became active and “extremely optimistic,” but also suffered insomnia, tension headaches, and visual hallucinations. Once her phenelzine was discontinued, her depression returned with no other signs or symptoms. She was subsequently restarted on phenelzine without the ginseng supplement but experienced no improvement in her depression at this time, leading to suspicion that the combination of phenelzine and ginseng may have enhanced the former’s psychoactive effects the first time it was prescribed [54]. The authors proposed that ginseng might inhibit cyclic adenosine monophosphate (cAMP) phosphodiesterase, leading to potential enhancements of the psychoactive effects of MAO inhibitors like phenelzine [14]. Little research about this specific drug–herb interaction has been conducted beyond this single case report.

### 5.4. Imatinib

*Panax ginseng* has demonstrated both therapeutic promise and potential interaction risk when used alongside imatinib in the treatment of chronic myelogenous leukemia. In vitro studies show that ginseng can enhance the anticancer effects of imatinib by attenuating the activation of signal transducer and activator of transcription 5 (STAT5) and p38 MAPK, key pathways associated with chemoresistance and cell survival [62]. This herb-drug combination led to increased apoptosis and suppression of anti-apoptotic proteins in human KBM-5 leukemia cells, suggesting a synergistic anticancer effect [62]. However, a case report describes a 26-year-old man who developed acute hepatotoxicity after consuming *Panax ginseng* energy drinks while taking long-term imatinib [35]. Liver function normalized after both agents were discontinued, and no further toxicity occurred when imatinib was reintroduced alone, implicating ginseng as a contributing factor [35]. The proposed mechanism involves ginseng-mediated inhibition of CYP3A4, the primary enzyme responsible for imatinib metabolism, potentially leading to elevated plasma levels and toxicity [35]. Together, these findings highlight that while ginseng may enhance imatinib’s therapeutic efficacy at the cellular level, it may also impair drug metabolism and increase the risk of adverse effects in clinical settings. Regardless, this specific interaction is scarcely researched outside of these two small studies, which, in tandem with the already unclear role of ginsenosides in CYP3A4 activity, makes more evidence necessary to make sufficiently definitive claims.

## 6. Discussion and Summary of Results from Human Experiments and Case Reports

A significant limitation of the current research on KRG is the small scale and low quantity of the human experimental studies that have been carried out, with most recruiting fewer than 30 participants. For example, Kim et al. (2022) studied 13 Korean men and demonstrated that the bioavailability of certain ginsenosides (Rg3, Rk1 + Rg5, F2, compound K) was improved when ginseng was administered in bioconverted form, while Lee et al. (2014) enrolled 25 men and reported reduced plasma ethanol levels and attenuated hangover symptoms after ginseng ingestion despite elevated acetaldehyde levels [10,23]. Yang et al. (2014) carried out a study with 22 young women, which found KRG reduced oxidative stress markers, improved constipation, and diminished menstrual pain and irregularity in its participants [42].

Other small trials have investigated drug–herb interactions. Yuan et al. (2004) worked with 20 healthy U.S. adults and found that American ginseng reduced the anticoagulative effect of warfarin, whereas Lee et al. (2008) (with 25 Korean stroke patients) reported no significant changes in warfarin’s activity, highlighting important inconsistencies in KRG’s interaction with the medication [43,44]. Further pharmacokinetic studies, such as Malati et al. (2012) (12 U.S. participants) and Kim et al. (2015) (14 Korean men), confirmed mild modulation of CYP450 activity but without strong clinical significance [52,53].

Taken together, these experiments suggest that ginseng may influence drug metabolism and exert antioxidative effects, but conclusions are constrained by small sample sizes, short durations, and variable ginseng formulations. In the absence of animal data, these findings would provide only tentative, low-certainty evidence for human benefit, heavily limited by lack of replication and absence of Phase II/III trials.

Clinical case reports and small series add further insight but underscore significant safety and heterogeneity concerns. Paik et al. (2015) and Norelli et al. (2015) document manic patients in individuals without prior history [34,40]. Jones et al. (1987) highlighted a case of overstimulation (including symptoms of insomnia, tension headaches, and visual hallucinations) in a patient consuming ginseng while on phenelzine, with resolution of symptoms upon discontinuation [54]. Paik et al. (2015) additionally highlights endocrine effects like gynecomastia in a boy and an adult man speculated to be linked to their recent ginseng consumption [34]. The report also highlights the Caron et al. (2001) study with 30 healthy young adults who experienced slight QTc prolongation and mildly reduced diastolic blood pressure with KRG administration [34]. Torbey et al. (2011) add further evidence for KRG’s capacity to induce QT prolongation with its report of a woman consuming high quantities of ginseng beverages for months who experienced QTc prolongation along with new symptoms of syncope and tonic clonic seizures [39]. Bilgi et al. (2010) discusses a man’s development of acute lobar hepatitis after drinking energy drinks with ginseng for 3 months, with liver enzymes following after cessation of ginseng; the man was also taking imatinib for chronic myelogenous leukemia, with authors speculating the presentation to be a result of KRG’s interaction with the imatinib-metabolizing CYP3A4 enzyme [35]. Janetzsky et al. (1997) documents a case of KRG’s potential interference with warfarin’s anticoagulative properties, contributing to the pile of conflicting evidence on the herb’s interaction with the medication [48].

Sample sizes in these case reports are uniformly small (*n* = 1–4), patient demographics vary widely, and confounding factors such as co-medications, age variations, and differences in lifestyle limit generalizability. Nevertheless, recurrent concerns do emerge across these case studies regarding potential psychiatric events, drug–herb interactions, and cardiovascular effects when ginseng is consumed in large amounts. If considered without the broader animal literature, these reports would suggest that while ginseng may exert biological activity relevant to human physiology, its clinical application is hindered by unpredictable responses and possible adverse events, warranting cautious interpretation and underscoring the need for systematic, controlled human trials

## 7. Conclusions

As it stands, *Panax ginseng* appears to offer promising therapeutic effects when it comes to the treatment of alcohol use disorder (AUD), according to various animal studies in the literature. However, one of the major limitations in the current body of evidence for KRG’s impact on AUD is the small number of available high-quality, large-scale human trials investigating their relationship, as discussed in the previous section. Much of the data found through this review was derived from in vitro laboratory investigations, preclinical animal studies, or small-scale clinical investigations. Furthermore, much of the evidence from human-based studies centers on hangover symptom reduction (e.g., Lee et al., 2014) rather than a more nuanced treatment of AUD, underscoring the absence of Phase II/III clinical trials necessary to more rigorously assess therapeutic efficacy in this population [23]. Beyond study size, scope, and quantity concerns, it is worth noting that many of these studies originate from South Korea, a nation with a significant market centered on ginseng. This introduces a potential conflict of interest that serves to further limit the applicability of results. Additionally, there has been very little exploration of sex-specific, age-related, and genetic factors that may influence KRG metabolism, efficacy, or safety.

Due to its relatively lax regulations as a supplement, *Panax ginseng* poses the risk of being a confounding factor during medical management of a wide variety of patients, particularly in vulnerable populations with hepatic, cardiovascular, psychiatric, and endocrine risk factors. Likewise, its potential interactions with CYP3A enzymes add further concern for its use in patients with AUD. However, the lack of standardized KRG formulations, wide variation in ginseng and ginsenoside dosages, and differences in experimental protocols occasionally often lead to contradictory findings, making it difficult to compare results across studies while making it difficult to provide an effective analysis of KRG’s safety profile for these populations. The inconsistent results in some of the human studies make it difficult for clinicians to have proper, data-driven conversations with patients about the benefits and risks of *Panax ginseng* while they are taking other, more conventional medications for their conditions. With the existing literature in mind, clinicians should monitor patients on *Panax ginseng* closely to ensure that any herb–drug interactions or otherwise negative side effects can be addressed quickly, while also allowing for these patients to have the potential positive effects that *Panax ginseng* is purported to provide.

With all these gaps in mind, there is a pressing need for more research into the specific mechanisms, drug interactions, and appropriate dosing regimens of KRG and its ginsenoside components. To address some of these gaps, future research should prioritize larger, randomized, placebo-controlled human trials that move beyond acute hangover mitigation to a more holistic evaluation of KRG’s efficacy in treating individuals diagnosed with AUD. Discerning KRG’s potential to reduce alcohol cravings, mitigate a wider range of withdrawal symptoms, improve mood patterns, and prevent relapses over longer periods of time with longitudinal follow-up would provide a more complete picture of KRG’s potential efficacy as a supplement for individuals with AUD. Comparing these outcomes to more standard AUD pharmacotherapies would also be helpful in clarifying the extent of KRG’s therapeutic potential. Monitoring for hepatotoxicity, cardiovascular changes, hormonal alterations, and mood irregularities in participants throughout these studies would be useful in better determining KRG’s side effect profile. Such research would benefit from the development of a more widely recognized, standardized KRG formulation with clearly defined ginsenoside profiles to improve reproducibility between studies. In addition, incorporating pharmacokinetic assessments in tandem with investigations of the clinical outcomes would help elucidate the role of individual variability (from differences in factors such as diet, gut microbiota composition, genetic expressions of metabolic enzymes, and medication regimens) in the overall therapeutic response. Finally, broader investigation into differential responses across sexes, age groups, genetic backgrounds, and common comorbidities, such as liver disease or depression, will be critical for clarifying KRG’s overall clinical utility in the AUD population.

Beyond the limitations of the collective evidence base, another major limitation of this review is its conduction in a narrative format rather than as a systematic review or meta-analysis. Consequently, our process of identifying and selecting literature for the review was not standardized with predefined inclusion or exclusion criteria. While we attempted to include the most relevant and representative studies, there is potential that selection bias remains in our assembled study set, as certain publications might have been unintentionally overlooked due to issues with study accessibility or visibility within the databases we utilized. In addition, there is an inherent risk of interpretive bias in how the evidence was synthesized. Thus, the conclusions drawn from this work should be interpreted as a broad synthesis of existing knowledge rather than as a definitive, systematically derived evaluation.

## Figures and Tables

**Figure 1 diseases-13-00285-f001:**
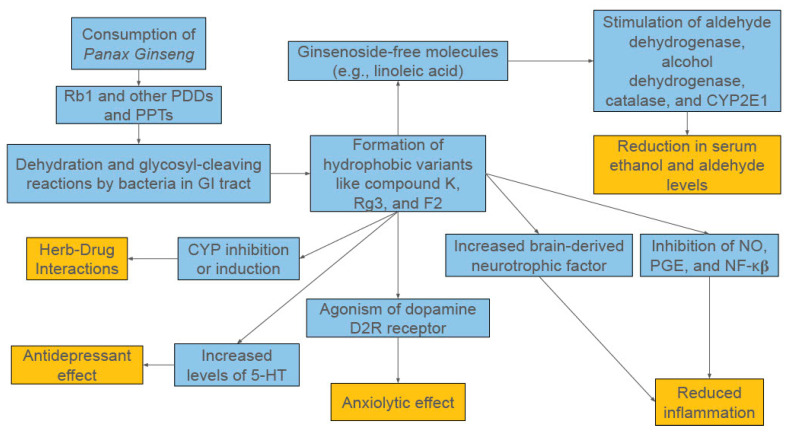
A Broad Overview of the Proposed Mechanisms of *Panax ginseng*.

**Figure 2 diseases-13-00285-f002:**
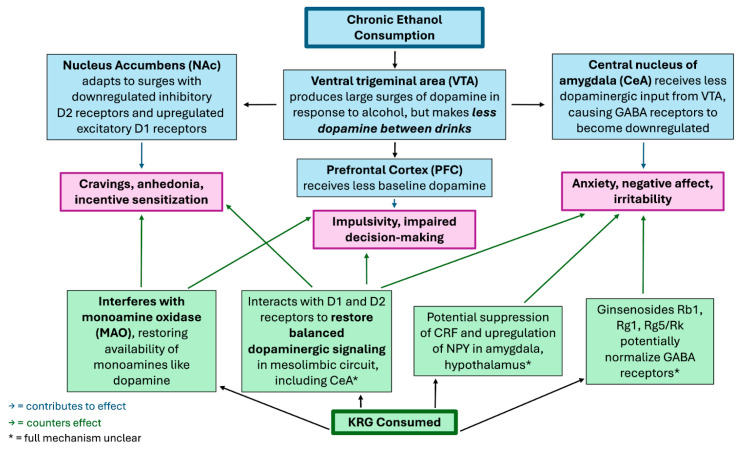
Speculated Mechanisms by Which Korean Red Ginseng (KRG) Modulates Neural Effects of Chronic Ethanol Consumption.

**Table 1 diseases-13-00285-t001:** Demographics and Major Findings of Animal and Human Studies of *Panax ginseng*.

Reference	Human or Animal Study	Sample Size	Country Study Conducted In	Major Findings
[22]-Kim et al.	Animal-mice	45–75 mice for each part of experiment	Republic of Korea	KRG alleviated withdrawal symptoms, improved spatial memory, lowered neuroinflammatory markers (TNF-α and NF-κβ), and restored BDNF signaling in a dose-dependent fashion, suggesting neuroprotective effects against alcohol-induced damage.
[10]-Kim et al.	Human	13 Korean men	Republic of Korea	Key ginsenosides Rg3, Rk1 + Rg5, F2, and compound K all showed higher oral bioavailability when KRG was given in its bioconverted form with hydrophobic ginsenosides removed (e.g., Rb1) compared to its usual extract form.
[23]-Lee et al.	Human	25 men	Republic of Korea	After consuming ethanol, men who consumed ginseng directly after had lower plasma ethanol concentrations in the initial hour and reported less subsequent hangover symptoms than the placebo group. Interestingly, ginseng group also had higher acetaldehyde levels despite this compound being associated with hangover symptoms, suggesting ginseng’s antioxidative effects may have masked the detrimental symptoms of higher acetaldehyde levels.
[29]-Zhao et al.	Animals—rats	4–6 rats for each part of experiment	Republic of Korea and China	Rats induced with ethanol withdrawal that received KRG expressed less anxious behavior than those that did not. A subsequent experiment showed this anxiolytic effect was blocked by a selective D2 receptor antagonist, but not a D1 receptor antagonist.
[31]-Lee et al.	Animals—rats	40 rats	Republic of Korea	Rats receiving ethanol and KRG showed dose-dependent lower liver weight, hepatic triglycerides, total cholesterol, and plasma triglycerides compared to ethanol-only group. Additionally, they had less hepatic inflammatory cells, less fat droplets, no necrotic cells, and reversed AMPK suppression, all of which suggest a protective effect for AFLD.
[18]-Jeon et al.	Animals—various	4 rats, 5 mice	Republic of Korea	Rats and mice were found to have differences in the ginsenosides they absorbed from oral KRG extract, with mice showing Rg3 and compound K in the plasma (akin to humans), while rats did not. Additionally, the species metabolized certain ginsenosides (i.e., Rb1, Rb2, Rc, and Rd) at different rates. Highlights the importance in species selection when extrapolating findings to human models.
[13]-Lee	Animals—rats	36 rats	Republic of Korea	Rats with induced morphine withdrawal that were given ginseng were found to have reduced anxious and depressed behavior using an elevated plus maze and forced swimming test. Changes in these rats’ CRF and neuropeptide Y expressions were identified in the hypothalamus when compared to controls.
[21]-Lee et al.	Animals—rats	32 rats	Republic of Korea, USA	Absorption of ethanol in the GI tract was found to be 21.0% less in rats that received KRG compared to controls, with little difference found when administered intraperitoneally. Demonstrates KRG lowers ethanol’s oral bioavailability when given concurrently.
[25]-Lee et al.	Animals—mice	20 mice	Republic of Korea	Serum ethanol and acetaldehyde levels of mice were reduced in groups that were administered non-ginsenoside compounds from steam-dried ginseng berries. Most abundant of these compounds included linoleate, palmitic acid, and linoleic acid.
[26]-Je et al.	Animals—mice		Republic of Korea	ALT and AST activity in mouse plasma, along with ethanol and acetaldehyde levels, were reduced in mice that received injections from ginseng sprouts prior to being given alcohol. This coincided with PCR determining higher expression of alcohol and aldehyde dehydrogenase genes in the mice that received ginseng sprouts. Finally, the study showed the ginseng sprouts had a strong antioxidative effect in vitro.
[28]-Cha et al.	Animals—mice	10–15 mice for each experiment	Republic of Korea	Elevated plus-maze model was used to examine anxiolytic effects of ginsenosides in mice. Found that those receiving ginsenosides Rb1, Rg1, and a Rg5/Rk showed less anxious behavior than those without. Anxiolytic mechanism of the compounds was hypothesized to be due to their interaction with GABA_A_ and GABA_B_ receptors, but this mechanism was not thoroughly explored in this study.
[30]-Bhattacharya et al.	Animals—various	58–88 rats or mice for the 5 experiments conducted	India	Red and white ginseng were given orally to mice and rats, with their behaviors in open-field and elevated plus-maze tests. Neither showed effects on first dose, but when given twice over 5 days, anxiolytic effects were comparable to that of mice given diazepam.
[32]-Churchill et al.	Animals—chicks	80 + 64 five-day-old male chicks	USA	64 five-day-old male chicks were injected with between 0.25 and 5.0 mg of ginsenoside Rb1. The chicks were found to have dose-dependent decreases in separation distress compared to the negative control chicks.
[33]-Xu et al.	Animals—various	10 mice or rats for each experiment	China	An intestinal metabolite of ginseng, 20(s)-protopanaxadiol (S111), was found to have antidepressant effects in mice and rats comparable in potency to fluoxetine. This was consistent with dissection results that showed increased levels of monoamines in the brains of groups receiving S111 through anti-reuptake effects.
[34]-Paik et al.	Human	4 case reports Double-blinded study with 30 young adults-13 men, 17 women	Republic of Korea	Documents two cases of mania: a 56-year-old woman with previously diagnosed major depressive disorder with psychotic symptoms on haloperidol and clomipramine, and the other a 26-year-old male with no psychiatric history, both of whom experienced a manic episode after starting to take ginseng capsules for 2 weeks and 2 months, respectively. Neither had any further instances of similar episodes with cessation of ginseng. Documents a pair of case reports describing potential ginseng-induced gynecomastia: one in a man who had ingested ginseng for a long period of time, and the other a 12 y/o boy who had experienced gynecomastia after consuming ginseng for a month. Gynecomastia resolved for the 12 y/o boy after discontinuation of ginseng. Documents a 2001 study [34] with 30 healthy young adults that found KRG administration slightly prolonged QTc interval (15 ms) compared to placebo group (3 ms), while also mildly reducing diastolic blood pressure.
[35]-Bilgi et al.	Human	1 case report	USA	26 y/o man taking imatinib for 7 years for chronic myelogenous leukemia presented with elevated ALT, AST, ALP, and total bilirubin along with acute lobular hepatitis on liver biopsy 3 months after starting to drink energy drinks containing *Panax ginseng*. Ginseng was discontinued with no subsequent liver enzyme elevations. Suspected to be a result of ginseng’s interaction with CYP3A4, which is essential for imatinib metabolism.
[36]-Zhou et al.	Animal	30 male rats	China	Extracts of white and red ginseng were injected into mice, with no significant EKG or obvious histological abnormalities being observed compared to controls. However, mice receiving ginseng did have comparatively high creatine kinases and decreased calcium ATPase activity, suggesting potential subclinical myocardial damage or calcium mishandling.
[37]-Parlakpinar et al.	Animal	40 male rats	Turkey	Rats given KRG extract were found to have decreased blood pressure, elevated troponin I and myoglobin levels, and signs of diastolic dysfunction in comparison to negative controls. Histopathologic changes suggestive of potential myocardial injury and altered Cu-Zn superoxide dismutase gene expression were identified in these rats as well.
[38]-Aravinthan et al.	Animal	45 guinea pigs	Republic of Korea	Following an hour of normothermic ischemia followed by 2 h of reperfusion, guinea pigs previously fed for two weeks with KRG were found in comparison to negative controls to have improved aortic flow, coronary flow, cardiac output, and left ventricular systolic pressure. Animals also were found to have less dramatic EKG changes, suppressed lactate dehydrogenase and cardiac troponin I levels, and lower oxidative stress markers.
[39]-Torbey et al.	Human	1 case report	USA	43 y/o woman consuming multiple caffeinated beverages along with 4 L of KRG beverages daily for at least six months presented with multiple instances of syncope, tonic clonic seizure, and prolonged QTc (up to 720 ms). No subsequent episodes occurred when the patient stopped consuming KRG. Suggests that prolonged use of KRG may pose cardiac risks when taken with other stimulants.
[40]-Norelli et al.	Human	2 case reports	USA	Documents two instances of patients experiencing manic episodes with no prior psychiatric history: one was a 23 y/o man who had been administered cannabis and unspecified “large quantities” of red ginseng every day for a month prior to his breakthrough manic episode; the other was a 79 y/o male who experienced a manic episode after he started consuming more than his baseline ginseng intake at around 3 to 4 “condensed drinks” per day. Symptoms of mania disappeared a few days after stopping ginseng intake.
[41]-Kabalak et al.	Human	1 case report	Turkey	39 y/o woman with history of smoking, coffee consumption, and frequent use of ginseng orally and topically developed menometrorrhagia and sinus tachycardia with atrial premature beats. Symptoms were resolved within two weeks of stopping consumption of all these things. Authors speculate that ginseng could have contributed to abnormal uterine bleeding and tachycardia.
[42]-Yang et al.	Human	22 women 21–30 y/o	Republic of Korea	KRG was found to reduce urinary levels of bisphenol A and malondialdehyde, an oxidative stress biomarker, in young women. Also improved constipation and menstrual pain/irregularity.
[43]-Yuan et al.	Human	20–9 men, 11 women	USA	Healthy adults part of a 4-week study received warfarin consecutively over 3 days at the start of week 1 and 4. Starting in week 2, half took American ginseng twice daily, and the others took a placebo. The group receiving ginseng exhibited significantly lower peak INR and plasma warfarin concentrations two weeks after starting ginseng, indicating that American ginseng reduced warfarin’s anticoagulant effect. While the authors suggested the ginsenosides may have increased warfarin’s metabolism or clearance through interactions with hepatic enzymes, no specific mechanism was explored or proposed.
[44]-Lee et al.	Human	25–13 men, 12 women	Republic of Korea	2-week study of 25 ischemic stroke patients receiving warfarin either with or without *Panax ginseng* showed no significant differences in INR, peak prothrombin time, or area under the curve between the two groups, even though all patients showed increased anticoagulation relative to baseline. This suggests that *Panax ginseng* may not consistently antagonize warfarin, or that its impact could depend on unclear patient-specific factors, ginseng species, preparation method, or dosing regimens.
[45]-Wang et al.	Animals—various	10 mice/6 rats per experiment	China	Ginsenoside Rb1 was found to improve mobility and reverse depressive behavior in rodents subjected to chronic stress, with neurochemical assays finding associated increases in brain levels of serotonin, norepinephrine, and dopamine. Suggests Rb1 may exert antidepressant effects through modulation of monoamine systems in the CNS.
[46]-Jang et al.	Animals—rats	24 rats per experiment	Republic of Korea	In tail suspension and forced swim tests, white ginseng alleviated immobility and depressive behaviors in rats, with associated decreases in serum corticosterone levels and increased hippocampal serotonin concentrations noted. Serotonin concentrations were even higher than the rats receiving fluoxetine as a positive control. Suggests white ginseng may have antidepressant effects through modulation of stress responses and serotonin activity.
[47]-Shin et al.	Animals—mice	4–6 mice per group, 5–9 groups per experiment	Republic of Korea	Mice induced with serotonin syndrome with 5-HT_2_A agonist showed improvement to symptoms with administration of ginsenoside Re. Researchers observed protein kinase C δ (PKCδ) inhibition in the groups receiving ginsenosides, suggesting PKCδ may be a therapeutic target for ginsenoside Re against serotonergic symptoms.
[48]-Janetzsky et al.	Human	1 case report	USA	47 y/o man on a stable warfarin regimen had his INR dropped from 3.0–4.0 to 1.5 within two weeks of starting *Panax ginseng*, without any other changes to his medications, lifestyle, or diet. After discontinuing ginseng, the patient’s INR returned to baseline, suggesting a reduction in warfarin efficacy potentially linked to ginseng intake, although the relevant mechanism was stated to be unclear at the time.
[49]-Tian et al.	Animals—rats	12 rats	China	Co-administration of *Panax notoginseng* and aspirin significantly increased plasma salicylate concentrations in rats, roughly doubling them compared to rats that received aspirin alone. In tandem with an in vitro MDCK-MDR1 cells showing increased apparent permeability, the findings suggest that *P. notoginseng* might facilitate GI absorption of aspirin.
[50]-Tian et al.	Animals—rats	12 rats	China	Co-administration of *Panax notoginseng* with aspirin showed increased absorption of ginsenosides Rg1, Rb1, Re, Rd, and notoginsenoside R1 in rats. Results of MDCK-MDR1 cell assays indicated that aspirin and salicylic acid disrupted tight junction proteins, which likely contributed to the improved absorption. This, in tandem with [49], suggests a mutual enhancement of absorption when ginsenosides and aspirin are used together.
[51]-Yang et al.	Animals—rats	6 rats	Taiwan	Selegiline’s bioavailability in rats was dropped from approximately 18% to 7.2% with low-dose ginseng but increased markedly to 29% with high-dose ginseng administration. The authors suspect these effects to be mediated by cytochrome P450 enzymes, particularly CYP3A4, CYP2B6, and CYP1A2, which metabolize selegiline into desmethyl-selegiline and L-methamphetamine.
[52]-Malati et al.	Human	12–8 men, 4 women	USA	Participants were given single doses of midazolam and fexofenadine before and after 28 days of receiving oral KRG. KRG treatment was found to reduce midazolam exposure, indicating CYP3A enzyme induction, while fexofenadine pharmacokinetics were scarcely altered, indicating a lack of P-glycoprotein interaction. Recommended to check if patients are on any CYP3A substrates before permitting ginseng use.
[53]-Kim et al.	Human	14 healthy men	Republic of Korea	14 men had plasma samples collected after being given a variety of medications, including caffeine, losartan, dextromethorphan, omeprazole, fexofenadine, and midazolam both before and after being supplemented with KRG for two weeks. Weak inhibition of CYP3C9 and CYP3A4, and weak induction of CYP2D6 were identified through pharmacokinetic changes, but none were clinically significant.
[54]-Jones et al.	Human	1 case report	Canada	42 y/o woman who regularly consumed ginseng experienced symptoms of overstimulation (i.e., insomnia, tension headaches, visual hallucinations) after being started on phenelzine for depression. When phenelzine was reintroduced without ginseng, depression did not improve, but overstimulation was not experienced. Authors propose ginseng may have enhanced phenelzine’s psychoactive effects through inhibition of cAMP phosphodiesterase, though evidence of this mechanism is limited.

**Table 2 diseases-13-00285-t002:** *Panax ginseng* Reported Side Effect Prevalence.

System	Reported Side Effect (Excluding Drug Interactions)	Evidence Type	Number of Cases (Patient Count) or Incidence Rate
Hepatic	Herbal-induced liver injury (HILI)-nonspecific to ginseng	Literature review [4,55,56]	China: Incidence of 6.38 per 100,000; Iceland: 3 per 100,000 [4,55] 12,068 cases worldwide across 80 publications [56]
Cardiovascular	Cardiac pathologies/myocardial damage (QT prolongation, tachycardia, ST depression, T wave inversion, AV block, troponin elevation, LDL elevation, diastolic dysfunction, HFpEF)	Animal study (rats) [37]	Dose dependent: 100 mg/kg *Panax ginseng* extract: Not specified 500 mg/kg *Panax ginseng* extract: Not specified
	QT prolongation	Review [34]	30
	Arrhythmia	Case study [39,41] Review [34]	3 [34,41]
	Elevated creatine kinase, increased cardiac contractility	Animal study [36]	Not specified
	TIA secondary to hypertensive crisis	Review [34]	1
	Hypotension	Review [34]	~13
GI	Morning diarrhea	Review [34]	~45
CNS/ Psychiatric	Manic psychosis	Case study [40]	2
	Manic episode	Review [34]	2
	Nervousness	Review [34]	~33
	Sleeplessness	Review [34]	~26
	Depression	Review [34]	~13
Reproductive/endocrine	Menometrorrhagia	Case study [41]	1
	Gynecomastia	Review [34]	2

## Data Availability

No new data was created or analyzed in this review. Data sharing is not applicable to this review.

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
