# Peer review of "A Review of the Mechanisms and Risks of Panax ginseng in the Treatment of Alcohol Use Disorder"

_diseases, 2025, doi:10.3390/diseases13090285_

Round 1
Reviewer 1 Report
Comments and Suggestions for Authors
The authors did a good job creating such a comprehensive review on Panax ginseng. One of the major downfalls of the manuscript is that it does not have the part describing how the literature included in the presentation is collected. Since addiction disorders are constantly mentioned and is a focus of this review more detailed explanation with a scheme of dopaminergic signalization, dysregulation and effects of KRG should be added to help the reader better understand the concept. The authors did write a significant section on side effects of this plant extract however I urge them to make an introduction stating (to the best of their knowledge) the exact number of reported cases of side effects reported in the literature (excluding drug interactions). This should be the number of in total cases and per system. Finally, please include the section on study limitations.
Reviewer 2 Report
Comments and Suggestions for Authors
- In the introduction, it has been mentioned that Rg3 is linked to anti-tumor effects, while Rb1 affects CNS, but the section muddles these into one statement. It is not distinguishable which mechanisms or compounds relate to which disease.
- The review states that hydrophobic metabolites mediate KRG’s effects but does not address inter-individual variability in gut microbiota-mediated bioconversion. This is critical for clinical translation, as microbiome differences may alter efficacy. Recommend discussing how factors such as antibiotics and diet might impact this conversion.
- “KRG may alleviate some symptoms of alcohol withdrawal, enhance cognitive performance, and attenuate anxiety.” No indication is present here whether this is via GABAergic, serotonergic, or dopaminergic pathways.
- “Endocrine effects such as gynecomastia may occur due to the phytoestrogenic properties of ginsenosides.” There is no citation given to any ERα/ERβ binding affinity assays, and no hormonal in vivo assay is presented.
- Emphasize the absence of Phase II/III trials and the overreliance on hangover studies (Lee et al., 2014) as proxies for AUD treatment.
Reviewer 3 Report
Comments and Suggestions for Authors
Comments and Suggestions for the Authors
This is a comprehensive and well-structured review article that addresses the potential role of Panax ginseng (Korean red ginseng) in the context of alcohol use disorder (AUD). The manuscript demonstrates a solid understanding of the pathophysiology of AUD and effectively synthesizes findings from both preclinical and limited clinical studies. The sections on mechanisms of action, therapeutic effects, and safety are particularly thorough, and the inclusion of drug interactions is highly relevant for clinical readers.
Strengths of the manuscript:
Very detailed and logically organized review, covering mechanistic, therapeutic, and safety aspects.
Extensive and up-to-date reference list including both animal and human studies.
Clear and scientifically sound language that is easy to follow.
The discussion of safety issues and potential drug interactions significantly increases the clinical value of the paper.
Areas for improvement:
While the review is comprehensive, the clinical relevance could be emphasized more clearly, as the current text is dominated by preclinical data.
Clearly distinguish preclinical (animal) from clinical studies and provide an evaluation of the strength and limitations of the cited evidence to enhance the scientific value of the review.
Regarding references to related and previous studies, the bibliography is extensive and covers both animal models and clinical research. However, it lacks a clear emphasis on the quality and hierarchy of the cited sources. The review would benefit from a shorter, more synthetic list of key studies in the clinical context.
Condense the overly detailed descriptions of molecular pathways (e.g., NF-κB, cytokine cascades) and focus on presenting the key clinical implications and main conclusions. Examples of sections that could be shortened include:
From line 91 – a very detailed description of the chemical structure and biotransformation of ginsenosides, which could be shortened to a few sentences with a reference to a table or review.
From line 116 – an extensive description of alcohol pharmacokinetics and enzyme activity (ADH, ALDH, CYP2E1). It can be shortened, keeping only information relevant to ginseng’s role.
From line 158 – a long, detailed description of cytokine and enzyme cascades (NF-κB, iNOS, COX-2). It can be shortened to a general mechanism description with appropriate references.
From line 434 – the detailed doses, ejection fraction changes, and QTc intervals in several animal models and a clinical case can be shortened to conclusions about potential cardiac risks and rare clinical cases.
From line 458 – detailed case descriptions (age, doses, liters of beverages) can be condensed and presented as a table or summarized collectively.
Consider adding a concise conclusion or graphical summary highlighting the clinical implications and research gaps.
Overall, this is a strong manuscript that makes a meaningful contribution to the literature on herbal supplements in AUD. With some refinement focused on clarity, critical appraisal, and clinical perspective, it could become an even more impactful review.
Reviewer 4 Report
Comments and Suggestions for Authors
The review is notable for exploring multiple mechanisms of KRG—antioxidant, anti-inflammatory, and neuromodulatory—while integrating findings from both animal models and limited human trials. It addresses potential benefits as well as risks, including drug interactions, and highlights KRG’s potential as a complementary treatment for AUD, an area that remains underexplored.
- I recommend addition of study limitations which may include these points:
Limited number of high-quality, large-scale human trials. Considerable variability in KRG formulations, dosages, and study protocols, making findings difficult to generalize. Incomplete data on long-term safety, particularly in vulnerable populations. Insufficient exploration of gender-specific and genetic variations in response to KRG.
- Also, I recommend the addition of the future research part such as testing KRG’s activity in different AUD stages (withdrawal, relapse prevention, cognitive recovery) and comparing KRG against standard treatments or placebo. Definition of optimal dosage, treatment duration, and preparation methods to ensure reproducibility. Monitoring for hepatotoxicity, cardiovascular changes, hormonal effects, and mood alterations in prolonged use. Examination of the effects in different age groups, sexes, genetic backgrounds, and comorbid conditions (e.g., liver disease, depression).
In Table 1: Please delete year of publications in the table.
Round 2
Reviewer 1 Report
Comments and Suggestions for Authors
Everything is in order.